# Students' Perception about Sustainability in the Engineering School of Bilbao (University of the Basque Country): Insertion Level and Importance

**Zaloa Aginako** [1,*] and **Teresa Guraya** [2]

1   Department of Electrical Engineering, Engineering School of Bilbao, University of the Basque Country, 48013 Bilbao, Spain

2   Department of Mining and Metallurgical Engineering and Materials Sciences, Engineering School of Bilbao, University of the Basque Country, 48013 Bilbao, Spain; teresa.guraya@ehu.eus

*   Correspondence: zaloa.aginako@ehu.eus

**Abstract:** Almost three decades have passed since the Rio declaration, and after numerous initiatives developed to include sustainability in higher education, with the support of Education for Sustainable Development, it is worth wondering at what point is the process of inserting sustainability in university degrees. To clarify this question, engineering students were inquired, at the University of the Basque Country (UPV/EHU), about their perception of the insertion-level of sustainability and the importance they give to it (in environmental, social, and economic dimensions). The novelty of this study lies in the use of a new questionnaire, based on the students' activity. The instrument was designed ad hoc and was previously validated for this study. The results indicate a low insertion level of sustainability in its three dimensions in three engineering degrees analysed. Nevertheless, the research also shows that the students give great importance to Sustainable Development (SD), either in academic, personal, or professional spheres. The low insertion level of SD and the high interest of students should be considered by the academic institution as an opportunity to deep in its holistic approach to promote the integration of SD in university curricula, not only in engineering degrees.

**Keywords:** sustainable development; sustainability insertion level; engineering education; students' activity



## 1. Introduction

Moves towards creating a sustainable future require people to fully understand the complexity of the world we live in and appreciate the need for more sustainable ways of living [1].

In this current context, the role of engineering students and future professionals is to solve global problems with solutions that respect the environment and social justice, within the limits of environmental sustainability, and without compromising future generations. [2]. All this requires, according to several authors [3–6], the need to include extensively Sustainable Development (SD) in engineering degrees in higher education.

The insertion of SD in engineering curricula has advanced from environmental inclusion to other domains such as economic and social, thanks to the promotion of professional associations such as the IEA (International Engineering Alliance), which include SD competencies in their assessment programs [7] setting a global trend. However, it has been also promoted from the academic spheres, thus for example, the CRUE (Conference of Spanish Rectors), in 2005, raised already the need to include sustainable development competencies in university degrees all along the curriculum [8]. Subsequently, and thanks to UNESCO initiatives and the SDGs definition in 2015, the models that only included sustainability in higher education in the teaching sphere are being progressively surpassed by holistic inclusion models, which aim to insert the SDGs and, therefore, sustainable development, in all university settings [9], ([10], p. 53), not forgetting teaching [11].

The model adopted at the UPV/EHU is precisely a holistic one [12], strategically materializing the integration of sustainability in an institutional agenda called EHUagenda 2030 [13]. Within the EHUagenda 2030, the inclusion of the SDGs in degrees' curricula is defined from several initiatives, all of them focused on enriching the previous teaching model of the institution called "iKD" (a student-centred learning model), which is now renamed "IKDi3", where learning is multiplied thanks to research and sustainability. Hence the term "i3", where the "i" letter means precisely sustainability in Basque, the local official language [12]. The initiatives established in the EHUagenda 2030 to include sustainability in the curricula focus on including common sustainability competencies at all degrees and developing those competences through active learning methodologies [12]. Extracurricular training activities are also promoted from the university, so that students can insert the SDGs in their Final Year Project (FYP), are trained in gender equality, or do development cooperation or service-learning activities. All of them, following the objectives of the EHUagenda 2030, seek to involve and activate students to promote behavioural changes regarding SD since the involvement of students is essential to achieve an authentic transformation in their SD behaviour, according to [1]. Active methodologies, meanwhile, provide positive value in this learning context, since they promote higher-order thinking skills and the involvement of students, both aspects necessary for education for sustainable development [14]. Furthermore, thanks to these methodologies, sustainability competencies are better developed [15].

Therefore, due to the context of the study, to collect students' perceptions, a novel questionnaire was designed based on students' activity and not on their knowledge about SD. Specifically, it is intended to know the students' vision of sustainability in the degrees, asking them about the activities they have carried out inside and outside the classroom. In those activities, there are recognized aspects of one of the three dimensions of sustainability: environmental, social, or economic. The instrument was designed and validated following the procedure described in [16].

Many questionnaires in the literature (Table 1) ask engineering students about their sustainability knowledge [17–23], or about their attitude [22,24], given importance [17,21] and even behaviour or action [22,23]. However, only a few try to find out, by surveying students, the level of insertion of sustainability in university engineering curricula [25,26]. In none of them are the aforementioned topics analysed from the point of view of students' activity, as in this questionnaire. It is understood that the activity-based approach of this work adjusts better to the current context, in which the inclusion of sustainability tries to obtain transformational results beyond knowledge acquisition about SD (sustainability literacy), a transformation that progresses from awareness to the consequent change towards sustainable ways of life from a social, economic, and ecological point of view [27].

**Table 1.** Main characteristics of instruments and studies in the literature.

| Study | Instrument | Population/Sample | Analyses Dimensions | Perspective |
|---|---|---|---|---|
| Sunthonkanokpong and Murphy (2019) | Scales created ad hoc for the particular context and conceptualised based on 17 SDGs | Bachelor of Science in Industrial Engineering Students (Thailand) $N_{polulation}$: 505 $N_{sample}$: 390 | Environmental; Social; Economical | Awareness Attitude Action |
| Jung, Park, and Ahn (2019) | Ad hoc created, but questions about objective knowledge are taken from USGBC | 2 USA regional universities Green buildings and sustainable construction degree N: 47 (50% population) Civil Engineering and Construction management degrees N: 48 (35% population) | Environmental; Sustainable Construction (SC) | Environmental concern; Sustainable construction knowledge (objective know and subjective knowledge); Sustainable behaviours as consumers |

**Table 1.** *Cont.*

| Study | Instrument | Population/Sample | Analyses Dimensions | Perspective |
|---|---|---|---|---|
| Akeel, Bell, Mitchell (2018) | SLT: Sustainability Literacy Test developed ad hoc Based on ASK and SULITEST | Nigerian engineers $N_{Students}$: 232 $N_{Educators}$: 84 $N_{Practitioners}$: 126 | Environmental; Economical; Social; Crosscutting issues | Awareness Literacy Self-Assessment of knowledge |
| Sanchez- Carradedo et al. (2018) | EDINSOST Based on degree's Sustainability competences Ad hoc. | Spanish computing engineers | Environmental; Economic; Social | Insertion level assessing sustainability skills acquisition in degrees |
| Tan, Udeaja, Babatunde, and Ekundayo (2017) | Ad hoc. Create survey | UK one university $N_{sample}$: 330 $N_{responders}$: 87 Mainly Engineering students (construction related cv) | Background knowledge and concept; Policies and regulation; Environmental issues; Social issues; Economic Issues; Technology and Innovation | Knowledge level to determinate Insertion of SD in construction cv; Importance level |
| Biasutti and Frate (2017) | The Attitudes Toward SD scale Ad hoc | | Environmental; Economic; Social; Education | Attitude toward SD |
| Zwickle, Koontz, Slagle, and Bruskotter (2013) | ASK: Assessing Sustainability Knowledge ad hoc environmental aspects (adapted from Coyle 2005) | Ohio University undergraduate students $N_{sample}$: 10,478 $N_{responders}$: 13,489 | Environmental; Economical; Social | Knowledge |
| Hovart, Stewart, and Shea (2013) | University of Maryland Sustainability Knowledge Assessment Created ad hoc | Students at University of Maryland $N_{sample}$: 9170 $N_{responders}$: 1436 | Environmental; Economical; Social | Sustainability knowledge |
| Azapagic, Perdan, and Shallcross (2005) | International Survey: Environment and Sustainable Development Created ad hoc. | Engineering students of 21 universities in 10 countries $N_{responders}$: 3134 | Environment and SD; 4 topics: env. Issues; env. Legislation; env. Tools, and SD. | Knowledge Importance Previous environmental SD education |

For this research it is important to know not only the instruments or studies characteristics but also the results obtained with those studies. The main results obtained with the abovementioned instruments/studies (Table 1), in case there are published, are shown in Table 2.

Being the context of the study a transformational approach of SD, the study presented below is not based on asking students of Electrical Engineering (EE), Industrial Electronic and Automatic Engineering (IEAE), and Mechanical Engineering (ME) of the UPV/EHU about their sustainability knowledge but about their sustainable activity. Which is, precisely, the parameter that can best show the change in teaching and at the university in general. To determine the inclusion-level of sustainability in the degrees, students are asked about the frequency in which they carry out 15 activities in the classroom, activities that involve working with elements of SDGs. Aside from knowing the importance they give to sustainability in their training, they are asked about the importance they give to the previous 15 activities. They are also asked about their activity outside the classroom

and the importance they give to sustainability in other spheres such as professional or personal, aspects that also have been widely analysed by other authors [28]. This way, this new instrument may be a relevant contribution to the field, assessing sustainability inclusion level in higher institutions that integrate sustainability with a holistic perspective in student-centred learning contexts, as in EHEA.

**Table 2.** Main results of the studies of Table 1.

| Study | Main Results |
|---|---|
| Sunthonkanokpong and Murphy (2019) | Low knowledge<br>High attitude and action<br>High difference attitude-awareness<br>Difference between years |
| Jung, Park, and Ahn (2019) | Not differences among groups in objective knowledge and better scores in concern, behaviour, and subjective knowledge among students without preparation on Sustainable Construction (SC); In general, Little knowledge on SC |
| Akeel, Bell, Mitchell (2018) | General Unawareness about UN Decade;<br>Strength in economic themes, least known about topics of crosscutting issues |
| Tan, Udeaja, Babatunde, and Ekundayo (2017) | High importance, low knowledge (basic), more knowledge environmental; knowledge increases with students' level; |
| Zwickle, Koontz, Slagle, and Bruskotter (2013) | Similar scores in environmental al economic domains, less on the social;<br>Significant differences between freshmen and juniors; highest scores aeronautical engineers. |
| Hovart, Stewart, and Shea (2013) | Not differences among educational levels, but differences among colleges lowest scores at engineering school;<br>Important differences between knowledge and environmental behaviours<br>High level of concern<br>Little SD knowledge and skills acquisition<br>3 courses are necessary for a significant change |
| Azapagic, Perdan, and Shallcross (2005) | Differences of knowledge across countries according to environmental problems of regions; knowledge not influenced by gender or level of study; knowledge gap in social of economic component of SD; low level of knowledge but high interest in SD. More Important for them personally and for them as engineers, but most important for future generations |

## 2. Materials and Methods

### 2.1. Participants

The questionnaire was sent to EE, IEAE, and ME students at the Engineering School of Bilbao (ESB) at the UPV/EHU. These degrees are structured in 4 annual courses, the subjects of the first two years and part of the fourth one are the same for the three degrees. Students acquire the specialization in electricity, electronics, or mechanics in the third course, with the specialization subjects of the degree, and in the fourth course with the elective subjects of the area. That is, the students of the three degrees share 62.5% of the ECTS in their training.

Given that the results are going to be partially analysed segregated by courses and degrees, below, the enrolled students' population is described according to the students' percentages distribution among mentioned groups.

By courses: first course 22% of enrolled students, second course 20%, third course 23%, and fourth course 35%. By degrees: EE degree 12% of enrolled students, IEAE degree 38%, and ME degree 50%.

### 2.2. The Instrument

It is a questionnaire created and validated specifically for this study. The validation process is detailed in [16]. The research questions pretended to answer with the instrument are:

- Do students know what Sustainable Development (SD) and Sustainable Development Goals (SDGs) are?
- Do students perceive the inclusion of the environmental, social, and economic dimensions of SD in their engineering studies? Are there differences in the insertion-level among the three dimensions? Between different degrees? Between courses?
- Do the students participate in SD training activities promoted by the university?
- What importance do students give to sustainable development in their personal, academic, and professional sphere?

The questionnaire has 46 items; all but two are closed-ended questions. Below there is a brief description of the instrument (for more details see Figure S1 in supplementary materials):

Part 1: Students are informed about the objectives of the research, about anonymous and voluntary participation, as well as the aggregate treatment of data and other ethical aspects of the research. In addition to informing the participants, it is about reducing the common method biases as stated in [29] "These procedures (anonymity and ask for honest responses) should reduce people's evaluation apprehension and make them less likely to edit their responses to be more socially desirable, lenient, acquiescent, and consistent with how they think the researcher wants then to respond". In this first part, students are also asked about the course and degree they are taking, to obtain demographic data from the sample.

Part 2: In this part, there are two questions to determine if students know the definitions of the SD and the SDGs. The purpose of these questions is not to know the level of literacy they have about sustainability as in other published studies [17–23] but know if SD and SDGs terms are familiar to them.

Part 3: Scales. There are two scales with 15 items (5 for each dimension) that represent 15 activities with some elements of the SDGs (see items in Tables 4 and 16). The activities are the same for both scales. With these 15 items, two scales have been composed; one is the *insertion-level scale*, and the other is the *importance scale*. Both are Likert scales with 5 levels. In the insertion scale, these five levels are *in no subject, in some subject, in many subjects, in most of the subjects*, and *in all of the subjects, and the scale of importance* varies from 1 to 5 where 1 is *not important*, and 5 is *very important*. For more details see Tables 4 and 16.

Part 4: There are two questions to ask students if they participate in activities that promote SD inside or outside the university. The answers are dichotomous (*Yes, No*). In addition, in two open questions, they are asked about the nature of these activities.

Part 5: Through two questions, it is intended to know the importance that students give to sustainable development in their private, professional, and social life, as well as the role that training that includes SD can play in their labour insertion. Both have four response levels: for the first question, not important, somewhat important, important, very important, DK/NA; and for the second, not positive at all, somewhat positive, quite positive, very positive, DK/NA.

In the design process of the instruments, some of the recommendations of Podsakoff et al. and Choi et al. [29,30] were considered, both in the drafting of the items (improving scale items) and in the structure of the questionnaire to avoid possible common method biases caused by wording and distribution of items.

### 2.3. Survey Administration Process

This study has the approval of the CEISH-UPV/EHU Ethics Committee for Research with Human Beings of the University of the Basque Country, as the intervention was carried out with students at that university (report M10-2020-030, Figure S2 in supplementary materials). According to the guidelines set by the ethics committee, the teachers participating in the research could not administrate the questionnaire, and teaching hours could not be used to take the survey. Therefore, a self-administered questionnaire was sent to the students by telematic means to be completed outside the classroom. Sixteen college lecturers helped the research team to administrate the survey after formally accepted their

voluntary participation in the study (by signing a form). The voluntary teachers included in the *news forum* of their virtual classrooms a letter addressed to students with the request to participate in the study and a link to the Google Forms questionnaire.

The first request was sent to students on 16 November 2020; the questionnaire remained open for three weeks, until 4 December. Two weeks after the first request (on November 30) and before closing the questionnaire, a reminder and a thankyou message were sent, following the same procedure, with the aim to get a higher response rate and reduce, therefore, the possible non-respondent bias. The survey was sent to all groups of the 3 degrees in a single subject. Taking into account different factors such as the low use of the virtual classroom of some students, the students that were only enrolled in their FYP, or the students enrolled in incomplete courses, it is estimated that the survey has reached 560 students. Once the questionnaire was closed, a database with the responses was prepared (for more details see Table S1 in supplementary materials). The data were processed with statistical analysis software.

### 2.4. Data Analysis

All quantitative data analyses were performed with the IBM SPSS V26 © program. In a first approach, an exploratory analysis of the items was made using descriptive statistics, histograms, and box-and-whisker plots, identifying two outliers that were eliminated for being invalid responses. In this first approach, it was also observed that all the answers' distributions were strongly asymmetric. The Shapiro–Wilk normality test was also performed for the items on the scales and for questions in part 5 of the questionnaire.

In addition, 6 new grouped variables were created to analyse the dimensions: three for the *insertion-level scale* and three for the *importance scale*. Each new variable represents the set of items of the social, environmental, and economic dimensions in the two studied scales. These grouped variables were calculated for each record (individual) as the mean value of the answers of each dimension's 5 items. The 6 grouped variables (dimensions) were used to analyse the differences between groups (degrees and courses). Cronbach's Alpha was calculated for each of these grouped variables in order to refute its internal consistency, already verified in the scale validation process [16].

After analysing the data globally, the grouped variables, or dimensions, were compared among groups (courses and degrees). These comparisons were made with the non-parametric Kruskal–Wallis test, since the normality condition of all the groups compared was not met ([31], p. 83) and because for non-symmetrical distributions the non-parametrical Kruskal–Wallis test results in a higher power compared to the classical one-way ANOVA [32]. Therefore, a significant value of *p* can be interpreted as a rejection of equality of medians [33].

In the cases in which the Kruskal–Wallis test showed a significant difference in medians, the effect size of the difference was calculated with the statistic $\eta^2$ [34]. Next, to determine among which pairs of groups were the differences observed with the Kruskal–Wallis test, the Mann–Whitney *U* post hoc test was performed between paired groups, with significance level adjustments by Bonferroni. Thus, it was possible to obtain paired comparisons between the answers of students of different degrees and courses and the differences' effect sizes in this test case using the *r* [34]. In the results section, the conducted tests are indicated in each analysis.

### 3. Results

#### 3.1. Participation

The number of responses obtained is shown in detail in Table 3; the valid number of responses per course and degree are indicated, and the percentages compared to the total number of valid responses (that is, 143).

**Table 3.** Participation rates.

|  |  | Degrees | | | Total/Course | |
| --- | --- | --- | --- | --- | --- | --- |
|  |  | **EE** | **IEAE** | **ME** | *No* | **%** |
| | 1st | 9 | 8 | 11 | 28 | 20% |
| | 2nd | 2 | 22 | 15 | 39 | 27% |
| Course | 3rd | 7 | 19 | 15 | 41 | 29% |
| | 4th | 2 | 21 | 12 | 35 | 24% |
| Total/Degrees | *No* | 20 | 70 | 53 | 143 | 100% |
| | % | 14% | 49% | 37% | | |

Given that the instrument was sent to 560 students and 143 answered (see Table 1), the participation rate was 25.9%, which can considerate an acceptable level of responses according to Nulty (2008) [35] that established the response level between 20% and 47% for self-administered online questionnaires. The table shows that the response percentage per course is similar, a little higher among third-year students. However, considering the% of students enrolled by courses, the highest proportion of respondent students are from 2nd grade (27% of respondents), 7 points up to the enrolled students' proportion in the second course (20%). The contrary happens with 4th-grade students with a response rate of 24%, lower than their proportion in the population (35%).

Taking into account the percentages of students enrolled by degree (12% EE, 38% IEAE, and 50% ME), the participation of the EEs is 2 points above their rate in the population; the IEAEs participate above their percentage in the population by 11 points, and MEs are underrepresented in almost 13 points.

The sample does not exactly match with the distribution of students by courses and degrees; however, being the sample small (143 registers), the ponderation of the sample was not considered [36].

### 3.2. Results, Knowledge

As indicated above, the aim of the study is not to do an exhaustive analysis of students' knowledge regarding SD and SDGs. The two questions related to the knowledge are to see if students are familiar with both terms and to observe if there are differences in the perceptions of two definitions, these data not being comparable with other studies [18–23] with in-depth analyses of the knowledge that students have about SD.

The results show that students have higher knowledge about sustainable development: 62% of students give a correct answer to SD definition, and only 2% choose the answer *DK/NA* (do not know/do not answer). However, when asked about the definition of SDGs, the results are considerably worse: only 16% give a correct answer, and 27% of students answer *DK/NA*; see Figure 1.

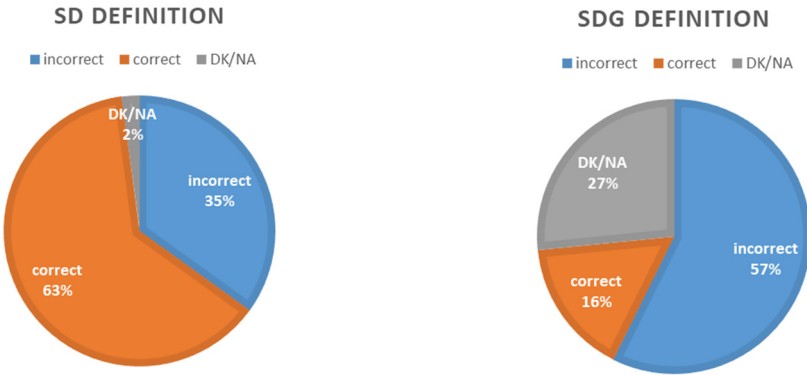

**Figure 1.** Answers' graphs for question 1 and question 2 of the questionnaire.

*3.3. Result, Insertion-Level*

In general, the low values of items shown in Table 4 indicate a low level of insertion of sustainability in engineering degrees. In all the items, approximately between 70 and 95% of the responses are grouped into the first three response levels of the scale (that is *in none, in some, and in many subjects*), and between 5 and 30% of the answers, at the high levels of the scale (*in most* and *in all subjects*). Given that no items' distribution is normal (strong positive asymmetry), median values will be taken into consideration to make comparisons; these values indicate that the central responses have been between *no subject* (1) and *some subject* (2). All indicate a low or very low level of insertion of SD in the three analysed degrees.

**Table 4.** Responses and some descriptive statistics of the insertion-level scale.

| Insertion-Level Scale | | | | | | | | | |
|---|---|---|---|---|---|---|---|---|---|
| **In the Activities that You Have Carried Out in Your Training during Your Engineering Courses, Either When Working on Theoretical Aspects, When Solving Problems, When Doing Projects or Internships or Seminars, Did You ...** | **Responses Level in (% of Responses)** | | | | | **Mean Value** | **SD** | **Median Value** | **Mode** |
| | **1** | **2** | **3** | **4** | **5** | | | | |
| **Environmental Dimension** ... analyse the impact of an adopted solution on biodiversity? | 44.8 | 49.0 | 4.2 | 1.4 | 0.7 | 1.64 | 0.70 | 2.00 | 2.00 |
| ... consider the complete lifecycle of elements, devices, or facilities? | 21.7 | 66.4 | 8.4 | 2.8 | 0.7 | 1.94 | 0.69 | 2.00 | 2.00 |
| ... consider as a design parameter to minimize the consumption of materials or resources? | 23.1 | 44.8 | 25.2 | 6.3 | 0.7 | 2.17 | 0.88 | 2.00 | 2.00 |
| ... identify measures to minimize contamination or damage in an environment? | 36.4 | 53.8 | 7.0 | 2.8 | 0 | 1.76 | 0.70 | 2.00 | 2.00 |
| ... assess that the desired solutions are energy efficient? | 32.9 | 35.0 | 24.5 | 7.0 | 0.7 | 2.08 | 0.96 | 2.00 | 2.00 |
| **Social Dimension** ... Identify the damages and/or benefits that the adopted solution will have for users? | 40.6 | 44.1 | 9.8 | 4.2 | 1.4 | 1.82 | 0.88 | 2.00 | 2.00 |
| ... identify the occupational hazards involved in certain projects or tasks? | 40.6 | 40.6 | 11.9 | 4.9 | 2.1 | 1.87 | 0.95 | 2.00 | 1.00 |
| ... assess the use of sensitive raw materials whose extraction harms specific populations, such as coltan? | 69.2 | 25.9 | 4.2 | 0.7 | 0 | 1.36 | 0.60 | 1.00 | 1.00 |
| ... consider the accessibility aspect to design friendly or ergonomic tools or solutions? | 58.7 | 32.2 | 7.0 | 2.1 | 0 | 1.52 | 0.72 | 1.00 | 1.00 |
| ... make decisions in accordance with the ethical principles of the profession? | 33.6 | 45.5 | 14.7 | 4.9 | 1.4 | 1.95 | 0.90 | 2.00 | 2.00 |
| **Economic Dimension** ... evaluate economic costs of a given solution in a comprehensive way? | 37.8 | 47.6 | 7.0 | 5.6 | 2.1 | 1.87 | 0.92 | 2.00 | 2.00 |
| ... critically analyse business actions, considering; for example, their impact on employment or social justice? | 52.4 | 41.3 | 4.2 | 2.1 | 0 | 1.56 | 0.68 | 1.00 | 1.00 |
| ... consider the economic viability of long-term solutions? | 51.0 | 37.1 | 8.4 | 3.5 | 0 | 1.64 | 0.78 | 1.00 | 1.00 |
| ... identify the social and environmental commitment of institutions and companies? | 54.5 | 40.6 | 4.9 | 0 | 0 | 1.50 | 0.59 | 1.00 | 1.00 |
| ... work in development cooperation scenarios, in international cooperation projects or at a local level? | 67.8 | 27.3 | 4.2 | 0.7 | 0 | 1.38 | 0.60 | 1.00 | 1.00 |

1: In no subject%; 2: In some subjects%; 3: Many subjects% (3); 4: Most of the subjects%; 5: In all subjects%.

Insertion-Level Scale Dimensions

The three grouped variables were used to make comparisons between courses and between degrees. The reliability of the grouped variables calculated with the five items of the dimensions were compared with the Cronbach's alpha test. For the three dimensions of the insertion scale, alpha values are higher than 0.7, which according to [37] is an acceptable value for internal consistency, as already published in the validity study of the scale [16]. The results are displayed in Table 5.

**Table 5.** Results of the reliability test, insertion-level scale dimensions.

| Insertion-Level Scale | Cronbach Alpha |
| --- | --- |
| Environmental Dimension | 0.787 |
| Social Dimension | 0.806 |
| Economic Dimension | 0.788 |

The grouped variables also present strong asymmetry in their distribution, as can be observed in the statistics values for these variables in Table 6.

**Table 6.** Descriptive statistics of the dimensions, insertion-level scale.

| Insertion-Level Scale | Mean Value | SD | Median Value | Mode |
| --- | --- | --- | --- | --- |
| Environmental Dimension | 1.92 | 0.58 | 2.00 | 2.00 |
| Social Dimension | 1.71 | 0.61 | 1.60 | 1.00 |
| Economic Dimension | 1.59 | 0.53 | 1.40 | 1.00 |

Regarding dimensions, in general, students perceive a higher insertion-level of the environmental dimension with a mean value for the dimension of 1.92 and a median of 2, followed by the social one (mean 1.71 and median value 1.6). The economic dimension has a lower presence, with a mean value of 1.59 and a median of 1.4. All of the values are between *in no subject* and *in some subjects*. To explore differences in the insertion-levels of each dimension among courses and among degrees, the grouped variables were used. The non-parametric Kruskal–Wallis test was conducted, and median values were compared since the grouped variables analysed do not have a normal distribution and are strongly asymmetric.

For the environmental dimension. The Kruskal–Wallis test output is in Table 7.

**Table 7.** Results, differences between courses, environmental dimension, insertion-level scale.

| | Groups | N | Mean | Median | Kruskal–Wallis H | p Value | Effect-Size $\eta^2$ |
| --- | --- | --- | --- | --- | --- | --- | --- |
| Environmental Dimension Insertion-level | 1st | 28 | 1.5643 | 1.6000 | 24.825 | 0.000 * | 0.157 |
| | 2nd | 39 | 1.8821 | 1.8000 | | | |
| | 3rd | 41 | 1.9171 | 2.000 | | | |
| | 4th | 35 | 2.2457 | 2.2000 | | | |

* Bilateral asymptotic significance, with significance level of 0.05. $\eta^2$ calculated with the tool in Psychometrica web site [38].

The test results indicate that there are significant differences among courses ($p < 0.05$) in the students' perception about the insertion-level in environmental dimension, with a large effect size: $\eta^2 > 0.014$ [39].

Given that there is a significant difference in the insertion level of the environmental dimension among courses and to determine what pairs of groups are different, post hoc tests were performed between pairs of groups. The used test was the Mann–Whitney's *U* test with a grade of significance adjusted by Bonferroni. The results of these tests are shown in Table 8.

**Table 8.** Post hoc tests, differences between courses, environmental dimension, insertion-level scale.

| Group by Group | Test Statistics | | Adjusted p Value [1] | Compared Groups Size | | Effect-Size $r = \frac{Z}{\sqrt{N}}$ | |
|---|---|---|---|---|---|---|---|
| | $U$ | $p$ Value | | $n_1$ | $n_2$ | | |
| 1st–2nd | 345.0 | 0.010 | 0.060 | 28 | 39 | - | |
| 1st–3rd | 340.0 | 0.004 | 0.023 * | 28 | 41 | 0.310 | medium |
| 1st–4th | 162.0 | 0.000 | 0.000 * | 28 | 35 | 0.403 | medium |
| 2nd–3rd | 744.5 | 0.594 | 3.564 | 39 | 41 | - | |
| 2nd–4th | 404 | 0.002 | 0.014 * | 39 | 35 | 0.164 | small |
| 3rd–4th | 471.5 | 0.010 | 0.058 | 41 | 35 | - | |

* Bilateral asymptotic significance, with significance level of 0.05. [1] Adjusted by Bonferroni.

The differences are between the first course and third course, first course and fourth course, and between the second course and fourth course, with effect-sizes between medium and small [39]. For the social dimension, the results of the Kruskal–Wallis test are shown in Table 9.

**Table 9.** Results, differences between courses, social dimension, insertion-level scale.

| | Groups | $N$ | Mean | Median | Kruskal–Wallis $H$ | $p$ Value | Effect-Size $\eta^2$ |
|---|---|---|---|---|---|---|---|
| Social Dimension Insertion-level | 1st | 28 | 1.5071 | 1.2000 | | | |
| | 2nd | 39 | 1.5949 | 1.4000 | 13.590 | 0.004 | 0.076 |
| | 3rd | 41 | 1.7659 | 1.8000 | | | |
| | 4th | 35 | 1.9200 | 1.8000 | | | |

$\eta^2$ calculated with the tool in Psychometrica web site [38].

The test results indicate that there are significant differences among courses ($p < 0.05$) in the students' perception about the insertion-level in social dimension, with a medium effect size: $\eta^2 > 0.06$. There are only differences between the 1st- and 4th-grade students, as can be seen in the results of the post hoc tests, Table 10.

**Table 10.** Post hoc tests, differences between courses, social dimension, insertion-level scale.

| Group by Group | Test Statistics | | Adjusted p Value [1] | Compared Groups Sizes | | Effect-Size $r = \frac{Z}{\sqrt{N}}$ | |
|---|---|---|---|---|---|---|---|
| | $U$ | $p$ Value | | $n_1$ | $n_2$ | | |
| 1st–2nd | 446.5 | 0.199 | 1.191 | 28 | 39 | - | |
| 1st–3rd | 376.0 | 0.014 | 0.087 | 28 | 41 | - | |
| 1st–4th | 260.5 | 0.001 | 0.008 * | 28 | 35 | 0.403 | medium |
| 2nd–3rd | 654.5 | 0.159 | 0.955 | 39 | 41 | - | |
| 2nd–4th | 442.5 | 0.009 | 0.053 | 39 | 35 | - | |
| 3rd–4th | 605.5 | 0.239 | 1.437 | 41 | 35 | - | |

* Bilateral asymptotic significance, with significance level of 0.05. [1] Adjusted by Bonferroni.

For the economic dimension in insertion-level scale, the results of the Kruskal–Wallis test are shown in Table 11.

**Table 11.** Results, differences between courses, economic dimension, insertion-level scale.

| | Groups | N | Mean | Median | Kruskal–Wallis H | p Value | Effect-Size $\eta^2$ |
|---|---|---|---|---|---|---|---|
| Economic Dimension Insertion-level | 1st | 28 | 1.4571 | 1.2000 | | | |
| | 2nd | 39 | 1.4974 | 1.4000 | 13.785 | 0.003 | 0.078 |
| | 3rd | 41 | 1.6049 | 1.4000 | | | |
| | 4th | 35 | 1.7829 | 1.8000 | | | |

$\eta^2$ calculated with the tool in Psychometrica web site [38].

The test results indicate that there are significant differences among courses ($p < 0.05$) in the students' perception about the insertion-level in economic dimension, with a medium effect size: $\eta^2 > 0.06$. There are only differences between the 1st- and 4th-grade students, and 2nd and 4th grades, as can be seen in the results of the post hoc tests, Table 12.

**Table 12.** Post hoc tests, differences between courses, economic dimension, insertion-level scale.

| Group by Group | Test Statistics | | Adjusted p Value [1] | Compared Groups Sizes | | Effect-Size $r = \frac{Z}{\sqrt{N}}$ | |
|---|---|---|---|---|---|---|---|
| | U | p Value | | $n_1$ | $n_2$ | | |
| 1st–2nd | 490.5 | 0.468 | 2.811 | 28 | 39 | - | |
| 1st–3rd | 422.5 | 0.060 | 0.362 | 28 | 41 | - | |
| 1st–4th | 276.5 | 0.003 | 0.017 * | 28 | 35 | 0.376 | medium |
| 2nd–3rd | 658.5 | 0.169 | 1.013 | 39 | 41 | - | |
| 2nd–4th | 406.5 | 0.003 | 0.015 * | 39 | 35 | 0.351 | medium |
| 3rd–4th | 527.5 | 0.045 | 0.271 | 41 | 35 | - | |

* Bilateral asymptotic significance, with significance level of 0.05. [1] Adjusted by Bonferroni.

No significant differences were found between degrees when comparing insertion-level scale dimensions with the Kruskal–Wallis test. Results can be seen in Tables 13–15.

**Table 13.** Results, differences between degrees, environmental dimension, insertion-level scale.

| | Groups | N | Mean | Median | Kruskal–Wallis H | p Value |
|---|---|---|---|---|---|---|
| Environmental Dimension Insertion-level | EE | 20 | 1.83 | 1.70 | | |
| | IEAE | 70 | 2.01 | 2.00 | 3.971 | 0.137 |
| | ME | 53 | 1.83 | 1.80 | | |

**Table 14.** Results, differences between degrees, social dimension, insertion-level scale.

| | Groups | N | Mean | Median | Kruskal–Wallis H | p Value |
|---|---|---|---|---|---|---|
| Social Dimension Insertion-level | EE | 20 | 1.45 | 1.20 | | |
| | IEAE | 70 | 1.77 | 1.70 | 5.489 | 0.064 |
| | ME | 53 | 1.71 | 1.60 | | |

**Table 15.** Results, differences between degrees, economic dimension, insertion-level scale.

| | Groups | N | Mean | Median | Kruskal–Wallis H | p Value |
|---|---|---|---|---|---|---|
| Economic Dimension Insertion-level | EE | 20 | 1.63 | 1.50 | | |
| | IEAE | 70 | 1.62 | 1.60 | 0.600 | 0.741 |
| | ME | 53 | 1.53 | 1.40 | | |

*3.4. Results, Importance*

3.4.1. Importance in Their Training

In the case of the importance scale, most of the responses focus on high response levels, concretely between 70 and 94% of the responses to the items are concentrated in response levels 4 and 5, while the response values from 1 to 3 only represent between 6 and 30% of the responses in each item. In addition, all the median values of the items are between 4 and 5, which indicates that students give high importance to activities of sustainability in their training. The results are detailed in Table 16.

**Table 16.** Responses and some descriptive statistics of the importance scale.

| Importance Scale | | | | | | | | | | |
|---|---|---|---|---|---|---|---|---|---|---|
| **Rate the Importance for Engineering Studies from 1 (Not Important) to 5 (Very Important) the Following Activities, to …** | | **Responses Level in (%)** | | | | | **Mean Value** | **SD** | **Median Value** | **Mode** |
| | | **1** | **2** | **3** | **4** | **5** | | | | |
| Environmental Dimension | … analyse the impact of an adopted solution on biodiversity. | 0.7 | 0 | 19.6 | 38.5 | 41.3 | 4.20 | 0.80 | 4.00 | 5.00 |
| | … consider the complete lifecycle of elements, devices or facilities. | 0 | 0.7 | 4.9 | 32.9 | 61.5 | 4.55 | 0.62 | 5.00 | 5.00 |
| | … consider as a design parameter to minimize the consumption of materials or resources. | 0.7 | 1.4 | 6.3 | 32.2 | 59.4 | 4.48 | 0.74 | 5.00 | 5.00 |
| | … identify measures to minimize contamination or damage in an environment. | 0 | 0 | 7.0 | 25.2 | 67.8 | 4.61 | 0.62 | 5.00 | 5.00 |
| | … assess that the desired solutions are energy efficient. | 0.7 | 1.4 | 4.2 | 24.5 | 69.2 | 4.60 | 0.70 | 5.00 | 5.00 |
| Social Dimension | … identify the damages and/or benefits that the adopted solution will have for users. | 0.7 | 0.7 | 9.8 | 32.9 | 55.9 | 4.43 | 0.76 | 5.00 | 5.00 |
| | … identify the occupational hazards involved in certain projects or tasks. | 1.4 | 3.5 | 9.1 | 21.7 | 64.3 | 4.44 | 0.90 | 5.00 | 5.00 |
| | … assess the use of sensitive raw materials whose extraction harms specific populations, such as coltan. | 2.1 | 4.2 | 12.6 | 32.2 | 49.0 | 4.22 | 0.97 | 4.00 | 5.00 |
| | … consider the accessibility aspect to design friendly or ergonomic tools or solutions. | 0.7 | 6.3 | 18.2 | 45.5 | 29.4 | 3.97 | 0.89 | 4.00 | 4.00 |
| | … make decisions in accordance with the ethical principles of the profession. | 1.4 | 2.1 | 16.1 | 32.9 | 47.6 | 4.23 | 0.89 | 4.00 | 5.00 |
| Economic Dimension | … evaluate economic costs of a given solution in a comprehensive way. | 1.4 | 2.8 | 9.8 | 39.2 | 46.9 | 4.27 | 0.86 | 4.00 | 5.00 |
| | … critically analyse business actions, considering, for example, their impact on employment or social justice. | 2.8 | 3.5 | 16.1 | 34.3 | 43.4 | 4.12 | 0.99 | 4.00 | 5.00 |
| | … consider the economic viability of long-term solutions. | 0.7 | 2.1 | 11.9 | 39.9 | 45.5 | 4.27 | 0.81 | 4.00 | 5.00 |
| | … identify the social and environmental commitment of institutions and companies. | 2.1 | 4.9 | 23.8 | 28.7 | 40.6 | 4.01 | 1.02 | 4.00 | 5.00 |
| | … work in development cooperation scenarios, in international cooperation projects, or at a local level. | 1.4 | 0.7 | 21.0 | 36.4 | 40.6 | 4.14 | 0.87 | 4.00 | 5.00 |

1: No important; 5: Very important.

The reliability of the grouped variables, or dimensions, was checked with the Cronbach's alpha test. Note that each dimension is formed by the mean values of five items (see Table 14). The alpha value is higher than 0.7 in all dimensions, which according to [37] is an acceptable value for internal consistency, as was reported in the scale validity study [16]. The alpha test results can be seen in Table 17.

**Table 17.** Results of the reliability test, importance scale dimensions.

| Importance Scale | Cronbach Alpha |
|---|---|
| Environmental Dimension | 0.781 |
| Social Dimension | 0.829 |
| Economic Dimension | 0.809 |

As shown in Table 18 the environmental dimension is perceived by students as the most important one (mean 4.49 and median 4.6) followed by the social dimension (with a mean value of 4.26 and a median of 4.4). The economic dimension is perceived as the least important for students (mean 4.16 and median 4.4).

**Table 18.** Descriptive statistics of the dimensions, importance scale.

| Importance Scale | Mean Value | SD | Median Value | Mode |
|---|---|---|---|---|
| Environmental Dimension | 4.49 | 0.51 | 4.60 | 5.00 |
| Social Dimension | 4.26 | 0.68 | 4.40 | 5.00 |
| Economic Dimension | 4.16 | 0.69 | 4.40 | 5.00 |

When comparing the perception of students in the three dimensions of importance scale between different degrees and courses using the Kruskal–Wallis test, there are only differences of medians between courses in the social dimension (Table 19).

**Table 19.** Results, differences between courses, social dimension, importance scale.

| | Groups | N | Mean | Median | Kruskal–Wallis H | p Value | Effect-Size $\eta^2$ |
|---|---|---|---|---|---|---|---|
| Social Dimension Importance | 1st | 28 | 4.29 | 4.40 | | | |
| | 2nd | 39 | 4.21 | 4.40 | 7.963 | 0.047 | 0.036 * |
| | 3rd | 41 | 4.10 | 4.20 | | | |
| | 4th | 35 | 4.46 | 4.60 | | | |

* Bilateral asymptotic significance, with significance level of 0.05. $\eta^2$ calculated with the tool in Psychometrica web site [38].

There is a Little difference among courses ($\eta^2$ = 0.036). To determine which groups are different, the Mann–Whitney $U$ test was conducted, the results of the test are in Table 20.

**Table 20.** Post hoc tests, differences between courses, social dimension, importance scale.

| Group by Group | Test Statistic | | Adjusted p Value [1] | Compared Groups Sizes | | Effect-Size $r = \frac{Z}{\sqrt{N}}$ |
|---|---|---|---|---|---|---|
| | U | p Value | | $n_1$ | $n_2$ | |
| 1st–2nd | 506.0 | 0.608 | 3.650 | 28 | 39 | - |
| 1st–3rd | 459.0 | 0.157 | 0.944 | 28 | 41 | - |
| 1st–4th | 427.0 | 0.378 | 2.268 | 28 | 35 | - |
| 2nd–3rd | 683.0 | 0.259 | 1.555 | 39 | 41 | - |
| 2nd–4th | 533.5 | 0.103 | 0.621 | 39 | 35 | - |
| 3rd–4th | 442.5 | 0.004 | 0.023 * | 41 | 35 | 0.331 | medium |

* Bilateral asymptotic significance, with significance level of 0.05. [1] Adjusted by Bonferroni.

The difference appears only between 3rd and 4th courses, with a medium effect size. In the cases of the environmental and economic dimensions, there are no differences as can be seen in Tables 21 and 22.

**Table 21.** Results, differences between courses, environmental dimension, importance scale.

| | Groups | N | Mean | Median | Kruskal–Wallis H | p Value |
|---|---|---|---|---|---|---|
| Environmental Dimension Importance | 1st | 28 | 4.44 | 4.60 | | |
| | 2nd | 39 | 4.51 | 4.60 | 0.880 | 0.830 |
| | 3rd | 41 | 4.46 | 4.60 | | |
| | 4th | 35 | 4.53 | 4.60 | | |

**Table 22.** Results, differences between courses, economic dimension, importance scale.

| | Groups | N | Mean | Median | Kruskal–Wallis H | p Value |
|---|---|---|---|---|---|---|
| Social Dimension Importance | 1st | 28 | 4.15 | 4.40 | 1.668 | 0.644 |
| | 2nd | 39 | 4.14 | 4.40 | | |
| | 3rd | 41 | 4.09 | 4.00 | | |
| | 4th | 35 | 4.27 | 4.40 | | |

Among degrees, there is no significant difference in the dimensions, as can be seen in the high *p* values of Tables 23–25.

**Table 23.** Results, differences between degrees, environmental dimension, importance scale.

| | Groups | N | Mean | Median | Kruskal–Wallis H | p Value |
|---|---|---|---|---|---|---|
| Environmental Dimension Importance | EE | 20 | 4.48 | 4.60 | 0.682 | 0.711 |
| | IEAE | 70 | 4.44 | 4.60 | | |
| | ME | 53 | 4.56 | 4.60 | | |

**Table 24.** Results, differences between degrees, social dimension, importance scale.

| | Groups | N | Mean | Median | Kruskal–Wallis H | p Value |
|---|---|---|---|---|---|---|
| Social Dimension Importance | EE | 20 | 4.10 | 4.20 | 3.917 | 0.141 |
| | IEAE | 70 | 4.19 | 4.40 | | |
| | ME | 53 | 4.41 | 4.40 | | |

**Table 25.** Results, differences between degrees, economic dimension, importance scale.

| | Groups | N | Mean | Median | Kruskal–Wallis H | p Value |
|---|---|---|---|---|---|---|
| Economic Dimension Importance | EE | 20 | 4.05 | 4.4 | 0.425 | 0.808 |
| | IEAE | 70 | 4.13 | 4.3 | | |
| | ME | 53 | 4.25 | 4.4 | | |

3.4.2. Importance in Personal Sphere

In the personal sphere and outside engineering training, students generally consider sustainability very important, with mean values between 2.8 and 3.5 (on a scale from 1: *not important*; to 4: *very important*). The results to this question are in Table 26.

**Table 26.** Results, importance of SD in the personal and professional spheres.

| Importance for . . . | Mean | SD | Median | Mode |
|---|---|---|---|---|
| You as Engineer | 3.50 | 0.648 | 4 | 4 |
| You as a Person | 3.48 | 0.636 | 4 | 4 |
| Future Generations | 3.48 | 0.812 | 4 | 4 |
| Society | 3.03 | 0.996 | 3 | 4 |
| Your Country | 3.01 | 0.978 | 3 | 4 |
| Your University | 2.80 | 0.933 | 3 | 3 |

Sustainability is more important for students in their professional sphere.

*3.5. Comparison of Insertion-Level and Importance Scales Responses*

The comparisons between the same items and grouped variables for the insertion-level scale and the importance scale are represented in Figure 2. The figure represents the

difference between the mean values, and the figures indicate how many times the mean value of the importance scale exceeds the mean value of the insertion-level scale for each item. In general, for all the items and dimensions, the mean values of the importance scale double or even triple the mean values of the insertion-level scale (both coded from 1 to 5).

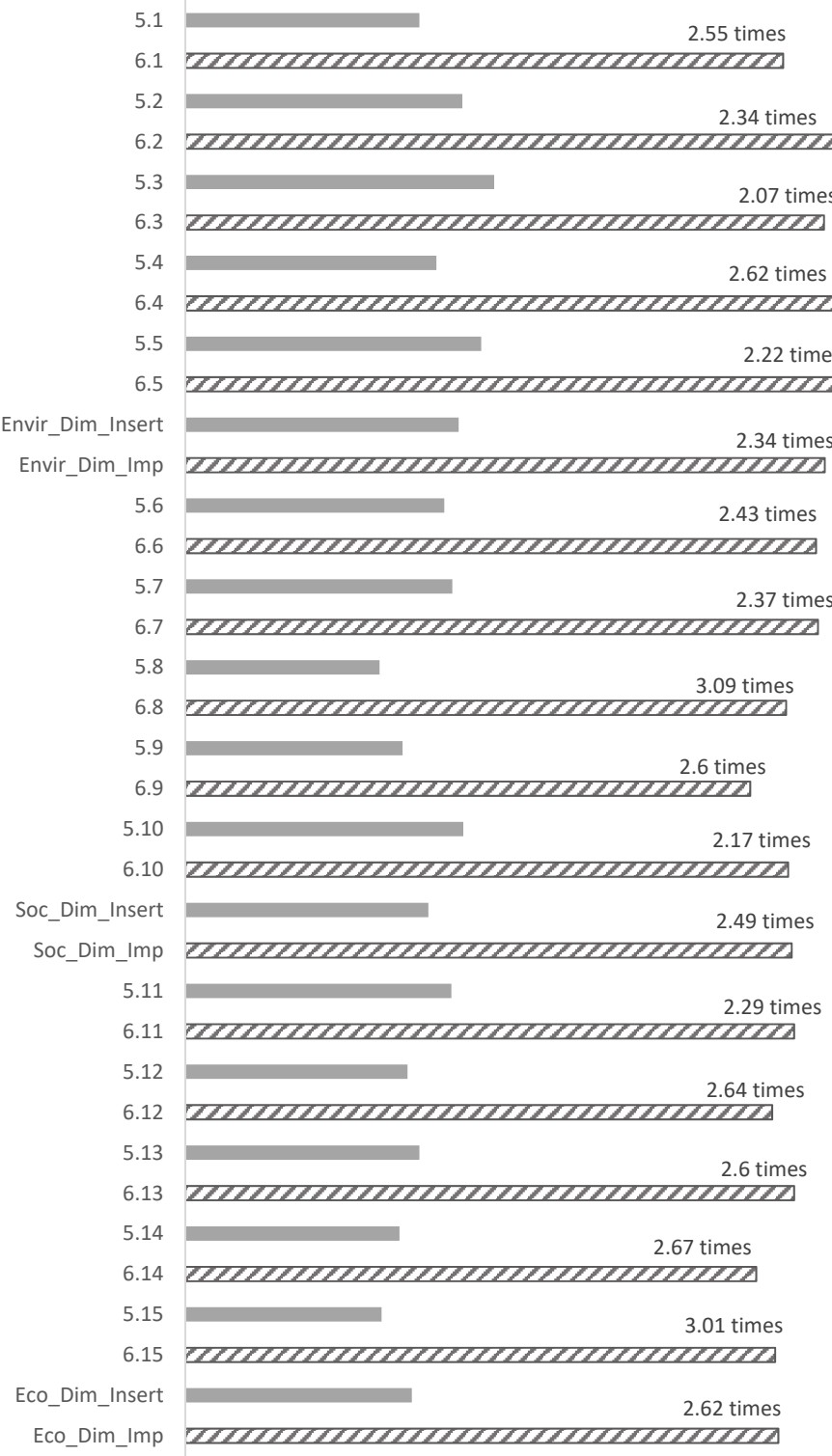

**Figure 2.** Comparison between insertion and importance scales' mean values.

### 3.6. Results, Participation in Sustainability-Related Programs at the University

The University of the Basque Country delivers courses and organizes activities to promote the SDGs among students, as already indicated above. With this question, we pretend to know the involvement level of the students in these university's programs that cover topics such as gender equality, SDGs inclusion in the FYPs, environmental footprint measurement, service-learning, etc. In general, students' participation in these institutional programs is practically null. Between 80 and 94% of students responded they did not participate in the institution's sustainability training programs. Nevertheless, their participation outside the university is higher since 31% of students say they have participated in activities related to sustainability outside the college.

In those cases in which students indicated the activities they did outside the university, the activities related to the environment stand out with 56% of the responses: cleaning beaches and rivers, planting trees, caring for species, work on maintaining forests, and Agenda 2021. The 41% refer to social volunteering activities, the majority with disadvantaged groups, in the Red Cross, with children, or in gender equality activities. Finally, only one student indicates having participated in an activity related to the economy, done within a student activity on the sustainable-energy economy.

### 3.7. Training on Sustainability and Labour Insertion

Students consider that the sustainability training received at the university will help them in their labour insertion, considering the training on sustainability, on a scale from 1 (*not positive at all*) to 4 (*very positive*), between *quite positive* and *very positive*: median 3 (mean value of 3.21). The results to this question can be seen in Table 27.

**Table 27.** Results, training on SD and labour insertion.

| | Responses Level in (%) | | | | Mean Value | SD | Median Value | Mode |
|---|---|---|---|---|---|---|---|---|
| | 1 | 2 | 3 | 4 | | | | |
| Do you think that incorporating sustainable development into your training would be a positive value to access the labour market? | 3 | 14.2 | 41.8 | 41 | 3.21 | 0.79 | 3.00 | 3.00 |

Responses levels: 1: not positive at all; 2: somewhat positive; 3: quite positive; 4: very positive.

## 4. Discussion

The insertion-level scale shows very reduced insertion values of sustainability in the analysed engineering degrees, with a little higher insertion-level in the environmental dimension, followed by the social and economic one. Similar results were found by [26] and [25], although the first research was not based on the students' activity but on their knowledge on SD, and the second on a survey on acquired sustainability competencies for students at Spanish universities where even authors state that sustainability competencies are not developed adequately. Lozano et al. [40] also found in a worldwide study a low insertion of SD through university programs and a little option to achieve an SD major at the bachelor's level, which seems to confirm that, in general, the insertion of sustainability is low in undergraduate levels.

An increase in the perception of insertion-level is seen as the courses increase; the same happens between knowledge and courses in [19,25,26]. In this research, the difference in favour of insertion-level in the environmental dimension in the 4th course may be attributed to the *environmental engineering subject* delivered in the 4th course. Nevertheless, in the study by Jung et al. [23], it is claimed that a single subject on SD is not enough to produce a difference in results or students' perception. Moreover, in another study [18], the authors advocate for at least three subjects in the curriculum to notice differences in knowledge of SD. In [16], a significant difference was registered in some items of this insertion scale, when applied to Environmental Engineering students, with a notable inclusion of sustainability competencies in the curriculum.

Regarding the values obtained in the importance scale, in general, engineering students of electrical, electronic, and mechanical degrees highly value the three dimensions of sustainability, in the following order of importance: environmental dimension, social dimension, and economic dimension. Similar results can be seen in the literature [17,20,26]. In addition, although most sources indicate that students prioritize the environmental aspect [17,28,41], there are also exceptions, such as in the study by [20] with greater knowledge of economic aspects or [19] with similar results for knowledge in the economic and environmental dimensions and less in the social one.

On the insertion-level scale, the surveyed students indicate a higher level of insertion of the environmental dimension and give to it greater importance on the importance scale. Other authors such as Fitzpatrick (2017) [3] also detected this imbalance and claimed the need to include the social and economic dimension with greater emphasis to turn engineers into change leaders.

The wide difference observed when comparing the insertion-level and importance scales (Figure 2) is remarkable. Something similar occurs in other studies [26,28] where high differences were found, in those cases, between students' low SD knowledge and the high importance they gave to it. The high importance that students give to SD within their academic sphere can be assumed as a great advantage to design approaches to promote the insertion of SD in the engineering syllabus, as claimed by some authors [17,20,26], who see in the importance that students give to SD, an opportunity to correct deficiencies in the integration of sustainability.

In the non-academic sphere of SD, in general, the results of this research are the inverse of the results of Azapagic et al. (2015) [21], where the same question was asked. In that case, students considered SD more important for entities or groups such as society or their country than to their personal or professional sphere, as is the case in this research; this difference may be attributed to cultural differences.

## 5. Conclusions

The most remarkable conclusion of this study is that sustainability is little or very little inserted in engineering in the EE, IEAE, and ME degrees of the UPV/EHU, when the analysis was done based on the activity of students, in the context of a holistic insertion approach of SDGs. In addition, students perceive, although with a small difference, that the environmental dimension is more inserted than the social or the economic dimension.

However, the students consider that the activities that include SD are highly important for their training. Notice that this perception extends to the three dimensions of sustainability and is not limited only to the environmental one.

They also consider that the inclusion of sustainability training in their degrees is a value between quite positive and very positive for their labour insertion.

Therefore, greater involvement of all the agents of the institution is required to incorporate SD in the curriculums. Thus, the authors of this work propose the following:

1.  It would be interesting also to know the teacher's point of view regarding these analysed issues. Therefore, an adapted version of this instrument could be administrated to lecturers too.
2.  When degrees' competences where defined, the SD competences had less visibility than today. The SD inclusion proven as a new necessity, on the part of the institution, it is necessary to work on the redefinition of some competencies of the degrees to promote the inclusion of SD.
3.  Finally, more training would be necessary to help teachers with the inclusion of sustainability competencies in the curricula and to systematize the evaluation of the insertion of these competencies in the degrees.

Finally, although it is true that the questionnaire has been designed for a specific context of engineering degrees, and therefore, the results cannot be directly extrapolated to other contexts, it is also true that the approach of the instrument based on students activity is valid in any of the areas of higher education.

*Limitations of the Study*

The design of this study responds to a practical approach, oriented to know the current baseline to establish actions to promote the insertion of SD in the analysed degrees. For that reason, the design itself supposes a limitation for the generalization of the results, being the results of this study limited to the case study. That is, to the three UPV/EHU engineering degrees analysed.

Another limitation of this study, due to the anonymous nature of responses, was that it was not possible to assess the aroused idea that non-responders could be students with little interest in the issue of SD. Generating a non-response bias that could modify the results of the section "students' interest on SD". To be able to carry on a possible non-response bias assessment with a later follow-up system, as proposed in [42], another survey administration procedure should be considered for future studies. For example, with non-anonymous responses, but with a commitment of not reveal the authorship of answers.

Finally, when doing conclusions, a new limitation was found, due to the design of the instrument. It was not possible to assess the dissonance between the high importance students gave to SD and their low participation in extracurricular activities at the UPV/EHU. A further study with focus groups should be done to clarify this point.

**Supplementary Materials:** The following are available online at https://www.mdpi.com/article/10.3390/su13158673/s1, Figure S1: Survey, Table S1: Data, Figure S2: Ethics committee approval certificate.

**Author Contributions:** Conceptualization, T.G.; Formal analysis, Z.A.; Funding acquisition, T.G.; Investigation, Z.A. and T.G.; Methodology, Z.A.; Writing—original draft, Z.A.; Writing—review & editing, T.G. Both authors have read and agreed to the published version of the manuscript.

**Funding:** This work has been carried out within the educational innovation project PIE 135/2019-2020, subsidized by the vice-chancellor's office for innovation, social commitment, and cultural action of UPV/EHU.

**Institutional Review Board Statement:** This study has the approval of the CEISH-UPV/EHU Ethics Committee for Research with Human Beings of the University of the Basque Country, report number: M10-2020-030.

**Informed Consent Statement:** Informed consent was obtained from all subjects involved in the study.

**Data Availability Statement:** Data are available in Table S1.

**Acknowledgments:** This work has been carried out within the educational innovation project PIE 135/2019-2020, subsidized by the vice-chancellor's office for innovation, social commitment, and cultural action of UPV/EHU. The authors thank the contributions of all teachers that helped to distribute the survey, and participant students. Finally, the authors would like to thank the CEISH of the UPV/EHU ethics committee, for their help in making the process respectful and in accordance with the ethical standards of this type of study.

**Conflicts of Interest:** The authors declare no conflict of interest. The funders had no role in the design of the study; in the collection, analyses, or interpretation of data; in the writing of the manuscript; or in the decision to publish the results.

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
