# Peer review of "Students’ Perception about Sustainability in the Engineering School of Bilbao (University of the Basque Country): Insertion Level and Importance"

_sustainability, doi:10.3390/su13158673_

Round 1
Reviewer 1 Report
The topic of the article is very interesting and current. The authors have made a very extensive review of the literature dealing with the Sustainable Development in engineering degrees in higher education. The study has an excellent research design, the authors described the implementation of the research adequately and the conclusions are very interesting. The article is very high quality.
Author Response
Thank you very much for your comments.
Reviewer 2 Report
The paper addresses the issue of sustainability insertion level into engineering degrees and the importance of sustainability issues for its students. The novelty of the approach is directly stated by the authors and is focused on the assessment of students' activity related to sustainability and its dimensions.
The paper is well written, referenced to actual and significant literature and has objectively important overall merit. The text has appropriate structure and covers it well with the content. The methodology or the paper is proper one for its objectives. The methodology used for research is very complex though but it is well explained and supplementary material provides appropriate explanations for its better understanding. The paper could significantly contribute to the development of sustainability perception among engineering students. The sample is limited to one university only but the questionnaire, the survey and complex handling of its results provide viable framework for research of sustainability perception among engineering students and attitude towards sustainability and its valuation.
I have some minor remarks to the text. The first one refers to one of the keywords, namely “insertion level”, which is much too general and should be referred to the sustainability issue (i.e. “sustainability insertion level”).
Second remark refers to the analysis of sustainability context, which is for most of the time considered as a set of three dimensions (economic, environmental and social), while as its core lies the integration of the dimensions. This is a common approach to the issue of sustainability and certainly it cannot be avoided within this paper since the questionnaire and the survey are already made and summarized. In my opinion, the mutual relationship between the different aspects of sustainability should be also assessed within this research but I don’t see enough space here for this in-depth interpretation in this paper specifically. Let’s treat this remark as a direction towards new research.
Author Response
Dear Reviewer,
Thank you very much for your comments, we have included the suggested keyword in the paper.
As you commented, the analysis of sustainability beyond dimension analysis is a good focus to be considered in future research. We preferred not conducted with the developed scales, because it is not recommended to use the same scales to do de-segregated analysis and overall analysis, due to a question of internal consistency, but of course, is a piece of interesting advice for future research, maybe including new questions in the instrument.
Best Regards
Zaloa Aginako
Reviewer 3 Report
I appreciate the opportunity to revise this paper. I think it deals with an important topic and is generally well written. I continue with some suggestions for the author(s):
- I would strongly recommend the authors to include a specific section devoted to literature review, presenting the main papers in the field related to this topic, but also including the arguments that could lead to proposing specific, testable hypotheses.
- To some extent related to my previous comment, what are the theoretical contributions/implications of the study. To which academic body of literature and/or theories you make a contribution and which one is this?
- Did you check any potential common method bias concern?
- Did you check any potential non-response bias? And early versus late respondent bias?
- I would also urge the authors to develop much more the limitations of the study, as well as the potential future avenue for further research that arise from their study.
- If possible, please avoid paragraphs including a single sentence.
I wish the authors luck in the project!
Author Response
Dear Reviewer,
We appreciate your suggestions very much, we tried to respond to all of them including some new content in the paper, as we consider the suggestions very pertinent to improve the quality of our paper.
Regarding the literature review, as the publication’s template does not indicate a specific section about the literature review as the reviewer suggests, we have included two tables ( 1 and 2) in the introduction with the main results of the literature review. Table 1 with the instrument/studies characteristics of the main paper analyzed to construct the instrument and design the study, and table 2 with the results of these studies, which leads to the justification of our work.
About theoretical contribution, we have included a little reference in lines 107-110. In the published validation of the instrument (Aginako et al. 2021), we included the theoretical contribution widely, this is the reason for not having included it in this new publication.
Regarding the common method bias, they were taken into account in the design phase of the survey. Thus, the recommendations of Podsakoff et al. (2003), Choi et al. (2010), and Azofra (2000) were considered in the drafting of the items and in the structure of the questionnaire to avoid possible biases caused by their effect. We are aware, however, that the survey is a bit long, although, it was necessary to include the questions needed to make an exhaustive analysis of the perception of students in all areas of interest. The way to reduce the bias of social desirability, lenient, and acquiescent was also taken into account in the administration of the survey, anonymity was guaranteed and the respondents were encouraged to answer honestly. With the disadvantage that non-responses, cannot be followed up as suggested by Sheridan and Strang (1998) with a subsequent telephone survey. Those considerations have been include in the paper in lines: 145-150 and 182-185
Regarding increasing responses to minimize non-response bias, a reminder was sent to students. Nevertheless, no further follow-up was possible with telephone calls or by email to non-respondents to value the possible non-response bias, because the responses were anonymous. These explanations have been included in the paper lines: 201-202
Although the idea that non-responders could be students with little interest in the issue who could modify the results of the section “students’ interest on SD” was valued, this theory could not be corroborated due to the anonymity of the responses. We did not make either a rigorous comparison between early responders and late responders to assess this possible bias since there is no certainty that the late responders represent non-responders, as partially occurs in the study on drug use by Studer et al. (2013). A limitation has been included in the paper to respond to this issue.
The section LIMITATIONS OF THE STUDY has been rewritten as the reviewer suggested.
We tried to avoid paragraphs including a single sentence.
We hope we have accomplished all the suggestions to consider this paper to be published. Thank you again for helping us to improve our work.
Round 2
Reviewer 3 Report
I have no further comments for the authors